# Association of short poor work ability measure with increased mortality risk: a prospective multicohort study

Marko Elovainio ,[1,2] Mikko Laaksonen,[3] Kainulainen Sakari,[4] Anna-Mari Aalto,[2] Tuija Jääskeläinen ,[2] Harri Rissanen,[2] Seppo Koskinen[2]

[1]Department of Psychology and Logopedics, University of Helsinki, Helsinki, Finland
[2]Finnish Institute for Health and Welfare, Helsinki, Uusimaa, Finland
[3]Finnish Centre for Pensions, Helsinki, Finland
[4]Diaconia University of Applied Sciences, Helsinki, Finland

**Correspondence to**
Dr Marko Elovainio;
marko.elovainio@helsinki.fi

## ABSTRACT

**Objectives** To examine whether a single-item measure of self-rated work ability predicts all-cause mortality in three large population-based samples collected in 1978–1980, 2000 and 2017.

**Setting** A representative sample of the population of Finland.

**Participants** The study population comprised 17 178 participants aged 18 to 65 from the population-based Mini-Finland, Health 2000 and FinHealth 2017 cohort studies, pooled together. In all cohorts, self-rated work ability was assessed at baseline (1978–80, 2000–2001 and 2017) using three response alternatives: completely fit (good work ability), partially disabled (limited work ability) and completely disabled (poor work ability) for work.

**Primary and secondary outcome measures** All-cause mortality from national registers. Cox proportional hazards models were adjusted for socioeconomic characteristics, lifestyle factors, self-rated health and mental health problems.

**Results** Of the participants, 2219 (13%) were classified as having limited and 991 (5.8%) poor work ability and 246 individuals died during the 4 year follow-up. The age- and sex-adjusted HR for mortality risk was 7.20 (95% CI 5.15 to 10.08) for participants with poor vs good work ability and 3.22 (95% CI 2.30 to 4.43) for participants with limited vs good work ability. The excess risk associated with poor work ability was seen in both genders, all age groups, across different educational levels, self-rated health levels and in those with and without mental health problems. The associations were robust to further adjustment for education, health behaviours, self-rated health and mental health problems. In the multivariable analyses, the HR for mortality among those with poor vs good work ability was 5.75 (95% CI 3.59 to 9.20).

**Conclusions** One-item poor self-rated work ability -measure is a strong predictor of increased risk of all-cause mortality and may be a useful survey-measure in predicting severe health outcomes in community-based surveys.

## INTRODUCTION

In the rapidly ageing working population, work ability is an important dimension of overall health and maintaining work ability is important for countries seeking to keep their citizens in work much longer than previous

## STRENGTHS AND LIMITATIONS OF THIS STUDY

⇒ Our observational study used three high-quality population-based data sets from three different historical contexts pooled together.
⇒ A wide range of important confounders including sociodemographic factors, lifestyle, health status and mental health problems .were considered in the analyses
⇒ The study is limited by being observational rather than interventional

generations.[1] The rapid changes in working life over the past few decades, with a shift from physical demands towards cognitive and psychosocial demands, together with changes in the functional capacity, expertise and attitudes of workers, is reflected in the content of work ability. Poor work ability represents a considerable challenge to the sufficiency of the workforce and to the sustainability of social protection schemes. Promoting the work ability of the working-age population helps people to maintain their health and functional capacity enabling longer working careers and may also improve the quality of life of the growing population of retirees.

There are efficient risk-adjusted, stepped care models and interventions for work disability management,[2] but these models and interventions need to be focused on those that really need and benefit from them. The primary precondition for correctly focused interventions is an easily administered and reliable measure of work disability. Currently, the identification of workers in increased risk of work disability is based on the duration of sickness absence.[3] Sickness absence is a good predictor of ill health and permanent work disability,[4] but only a small proportion of those with even long-term sickness absences really need sustained support to maintain work ability.[5] Thus, there is a clear need for a simple, easy to administer and unambiguous measures of work-related health and

**Table 1** Characteristics of the participants in the combined data

| | Total (N=17 176) |
|---|---|
| **Sex** | |
| Male | 8277 (48.2%) |
| Female | 8899 (51.8%) |
| **Age (years)** | |
| Mean (SD) | 44.0 (11.9) |
| **Education** | |
| Lower | 5800 (37.1%) |
| Intermediate | 5271 (33.7%) |
| Higher | 4559 (29.2%) |
| **Alcohol consumption** | |
| Low | 13 811 (88.0%) |
| High | 1885 (12.0%) |
| **Current smoker** | |
| No | 9191 (61.4%) |
| Yes | 5778 (38.6%) |
| **Body mass index** | |
| Mean (SD) | 26.1 (4.59) |
| **Mental health problems** | |
| No | 13 761 (89.2%) |
| Yes | 1660 (10.8%) |
| **Moderate or poor self-rated health** | |
| No | 10 928 (64.7%) |
| Yes | 5975 (35.3%) |
| **Follow-up time from baseline (years)** | |
| Mean (SD) | 3.91 (0.293) |

functional ability that can predict severe health outcomes and health service use. Such measures are also essential for health surveys and population health monitoring.

One of the most widely used ones is the Work Ability Index, a self-report survey screening instrument.[6 7] The questionnaire assesses with 7 dimensions and 16 questions the degree to which workers consider their state of health adequate to cope with their job demands. The continuous scores can be categorised into groups that reflect different levels of need of support. Although widely used and reliable,[1] the Work Ability Index is still rather complicated and inconvenient for use in large-scale surveys.[8] A more simple, single-item question on self-rated health (SRH) has been commonly used as a measure of health status in psychological research, clinical settings and general population surveys. It has also repeatedly been shown that SRH associates with physicians' assessments of health,[9 10] chronic disease incidence, physical and cognitive functional limitations, health services use, clinical biomarkers[11–14] and even mortality.[15–17]

In this study, we tested a self-administered one-item measure of work ability as a predictor of overall mortality.

The subjective experience of work ability has the advantage over general health evaluation that it has a clear point of reference: subjective work ability is a measure of perceived balance of the demands of the work and the resources of the individual.[18] Although the conceptual framework is, of course, much more complicated, the evaluation of one's capacity to cope with the demands of one's work is probably much less abstract and cognitively challenging than evaluating the overall health status. We tested whether subjective evaluation of work ability predicts the risk of overall mortality in three large and nationally representative cohorts collected in the 1970s, 2000s and 2010s. We also tested whether the one-item work ability measure predict mortality risk as well as and independently of the one-item SRH measure.

## METHODS

### The study population

The participants were from three population-based, nationally representative health examination studies: Mini-Finland Health Survey, Health 2000 and FinHealth 2017.[19] In the Mini-Finland Health Survey, 8000 persons (3637 men and 4363 women) aged 30 or over were invited to participate between 1978 and 1980.[20] The sample was representative of the Finnish population (response rate 90%) and the sampling design was stratified two-stage cluster sampling. The survey was carried out in 40 study areas around the country. The final sample in this study was 5897, after excluding persons aged 65 or over. The Health 2000 survey was carried out in 2000–2001. In total, 8028 Finns over the age of 30 and 1894 aged 18–29 years were invited to participate. Information about the participants was collected through interviews, an extensive health examination and several questionnaires. A shortened version of the study protocol was used in the age group 18–29 years.[21] Two-stage cluster sampling included 15 largest towns and 65 health districts in Finland. The response rate was 85% and the final sample used in this study included 6723 persons aged less than 65 years. The FinHealth 2017 health examination study was carried out in 50 localities in 2017, with the objective of evaluating 10 000 randomly selected persons aged over 18 years in Finland. The study consisted of a physical examination and questionnaires.[22] In total, 7055 people participated, yielding a response rate of 69% and the final sample used in this study covered 4556 persons aged 18–64 years. For the main analyses we combined these three data sets. We restricted the follow-up time to a maximum of 4 years, because that was the maximum follow-up time in the latest (FinHealth 2017) cohort to make data sets comparable. A total of 17 178 participants alive on the first day of the first phase of each survey were included. Characteristics of the participants in each individual data set are presented in online supplemental tables 1–3. All three surveys were conducted according to the Declaration of Helsinki. The design, population, and protocol of the individual cohorts have been described in detail elsewhere.[19 20 22 23]

**Table 2** Associations between poor work ability and mortality in subgroups. Numbers are HRs, 95% CIs and p values

| Study population | Work ability | N | Number of deaths | HR | 95% CI | P value |
|---|---|---|---|---|---|---|
| All | Limited | 16901 | 246 | 4.58 | 3.35 to 6.26 | <0.001 |
| | Poor | | | 12.41 | 9.21 to 16.72 | <0.001 |
| Sex | | | | | | |
| Men | Limited | 8140 | 186 | 4.92 | 3.45 to 7.03 | <0.001 |
| | Poor | | | 9.74 | 6.9 to 13.76 | <0.001 |
| Women | Limited | 8761 | 60 | 4.36 | 2.26 to 8.4 | <0.001 |
| | Poor | | | 19.77 | 10.89 to 35.9 | <0.001 |
| Age-group | | | | | | |
| Under 35 years | Limited | 3270 | 13 | 6.5 | 1.76 to 24.06 | 0.01 |
| | Poor | | | 12.29 | 1.61 to 93.97 | 0.02 |
| 35–44 years | Limited | 4292 | 28 | 4.4 | 1.72 to 11.22 | <0.001 |
| | Poor | | | 9.25 | 3.39 to 25.27 | <0.001 |
| 45–54 years | Limited | 4430 | 85 | 3.22 | 1.87 to 5.55 | <0.001 |
| | Poor | | | 11.1 | 6.78 to 18.16 | <0.001 |
| 55–64 years | Limited | 3870 | 118 | 2.34 | 1.46 to 3.73 | <0.001 |
| | Poor | | | 5.29 | 3.38 to 8.28 | <0.001 |
| Educational group | | | | | | |
| Lower | Limited | 5628 | 148 | 3.77 | 2.49 to 5.7 | <0.001 |
| | Poor | | | 8.19 | 5.5 to 12.21 | <0.001 |
| Intermediate | Limited | 5210 | 68 | 3.7 | 2.00 to 6.84 | <0.001 |
| | Poor | | | 13.33 | 7.57 to 23.45 | <0.001 |
| Higher | Limited | 4542 | 26 | 3.05 | 1.03 to 9.01 | 0.04 |
| | Poor | | | 10.05 | 3.34 to 30.29 | <0.001 |
| Self-rated health | | | | | | |
| Good | Limited | 11179 | 84 | 5.23 | 2.88 to 9.47 | <0.001 |
| | Poor | | | 15.72 | 7.95 to 31.08 | <0.001 |
| Moderate or poor | Limited | 5699 | 160 | 2.99 | 1.95 to 4.59 | <0.001 |
| | Poor | | | 7.93 | 5.27 to 11.92 | <0.001 |
| Mental health problems | | | | | | |
| Yes | Limited | 13730 | 172 | 4.89 | 3.4 to 7.05 | <0.001 |
| | Poor | | | 13.41 | 9.29 to 19.34 | <0.001 |
| No | Limited | 1655 | 34 | 3.1 | 1.28 to 7.5 | 0.01 |
| | Poor | | | 6.8 | 2.93 to 15.76 | <0.001 |

## Work ability

In all cohorts, self-rated work ability was assessed at baseline (1978–1980, 2000–2001 and 2017) using a single question: 'Regardless of whether you are employed or not, please estimate your current work capacity. Are you?' The response alternatives were 'completely fit for work', 'partially unable to work' and 'completely unable to work'. These categories will be referred to as good, limited and poor, respectively.

## Mortality

Follow-up for all deaths irrespective of the cause started at inclusion in the study cohort. Information on deaths and the dates of death were obtained from the Causes of Death registry maintained by Statistics Finland and the Death registry from the Digital and population data services agency until the end of 2020. The Causes of Death registry includes information of all deaths in Finland.

## Assessment of confounders

Potential confounders included common risk factors for poor work ability and mortality. In all subcohorts, we assessed sex, age, educational attainment (low, intermediate, high), cigarette smoking (current smoker (yes/no)), alcohol intake frequency (in FinHealth 2017 based on Alcohol Use Disorders Identification Test (low or intermediate vs high). In Mini-Finland and Health 2000 based on grams of absolute alcohol from detailed consumption

**Table 3** Associations between poor work ability and mortality (mean follow-up 3.9 years in the combined data set). Numbers are HRs, 95% CIs and p values

| Adjusted for (in addition to age and sex) | | | | | | |
|---|---|---|---|---|---|---|
| Models | Work ability group | N | No of deaths | HR | 95% CI | P value |
| None | Limited | 15 862 | 244 | 3.22 | 2.30 to 4.53 | <0.001 |
| | Poor | | | 7.20 | 5.15 to 10.08 | <0.001 |
| Cohort | Limited | 15 862 | 244 | 3.13 | 2.22 to 4.41 | <0.001 |
| | Poor | | | 6.50 | 4.56 to 9.25 | <0.001 |
| Education | Limited | 15 253 | 242 | 2.96 | 2.09 to 4.18 | <0.001 |
| | Poor | | | 6.35 | 4.48 to 9.01 | <0.001 |
| Health behaviours | Limited | 13 071 | 224 | 2.97 | 2.07 to 4.26 | <0.001 |
| | Poor | | | 6.04 | 4.18 to 8.72 | <0.001 |
| Self-rated health | Limited | 15 230 | 240 | 2.89 | 1.97 to 4.23 | <0.001 |
| | Poor | | | 6.17 | 4.15 to 9.16 | <0.001 |
| Mental health problems | Limited | 13 974 | 202 | 3.05 | 2.11 to 4.42 | <0.001 |
| | Poor | | | 6.19 | 4.11 to 9.31 | <0.001 |
| All above | Limited | 11 837 | 186 | 3.27 | 2.12 to 5.05 | <0.001 |
| | Poor | | | 5.75 | 3.59 to 9.20 | <0.001 |

reporting and divided to low and high (in men >280 g/week and in women >210 g/week),[24] body mass index (BMI ($kg/m^2$), obesity BMI ≥30), SRH (good or rather good vs moderate, rather poor or poor) and mental health problems (In Mini-Finland self-reported mental health problems (No vs Yes), in Health 2000 based on General Health Questionnaire-12[25] (classified as low and high with a cut-off of 4 points) and in FinHealth 2017 based on the 5-item mental health dimension of SF-36 (MHI-5)[26] (classified as low and high with a cut-off of 52 points)).

## Statistical methods

We used Cox proportional hazards models taking into account the survey design to estimate HRs for the association of work ability with all-cause mortality. Participants that were 65 years old or older (official retirement age) at baseline were excluded. The follow-up lasted from the study entry of each individual to death or end of the follow-up, whichever came first. The proportional hazards assumption was examined using scaled Schoenfeld residuals.

In the primary analyses, we examined whether having limited or poor work ability predicted death with the following steps. First, we assessed the unadjusted associations separately for men and women, for separate age groups, for separate educational levels and among those with and without poor SRH, and with or without mental health problems. Second, we assessed the associations of poor work ability with mortality risk adjusting for (1) age and sex, (2) age, sex and education, (3) age, sex, alcohol consumption, smoking status, BMI and physical activity, (4) age, sex and SRH, (5) age, sex and mental health problems and (6) all confounders. The percentage of excess risk explained (PERM) by the confounders was

calculated to assess the extent to which the associations of poor work ability with mortality were attributable to differences between individuals with poor work ability and the other individuals in the level of confounders included in the models.

$$PERM = ((HR_{\text{(age and sex adjusted)}} - HR_{\text{(age and sex + X adjusted)}})/(HR_{\text{(age and sex adjusted)}} - 1)) \times 100.$$

We also reported the time-dependent receiver operating characteristic (ROC) curve and area under the ROC (AUC) / for censored survival data for the age and sex adjusted model and then for the model that included age, sex and work ability by using the nearest neighbour estimator for the bivariate distribution function of ($X$, $T$), where $T$ represents survival time.[27] Finally, we calculated the net reclassification index (NRI) for each variable added to the age and sex adjusted models to evaluate the predictive value of work ability compared with all the other included variables. We conducted all data analyses in R (V.4.1.1).

## Role of the funding source

The funders of the study had no role in study design, data collection, data analysis, data interpretation or writing of the report.

## Patient and public involvement

No patient involved.

## RESULTS

The primary analysis in the combined cohort contained 17 176 participants (8899 women, 51.8% and 8277 men, 48.2%). The mean age was 44 years (SD=11.9 years). Of the participants, 37.1% had lower, 33.7% intermediate and 29.2% higher educational attainment. Intermediate

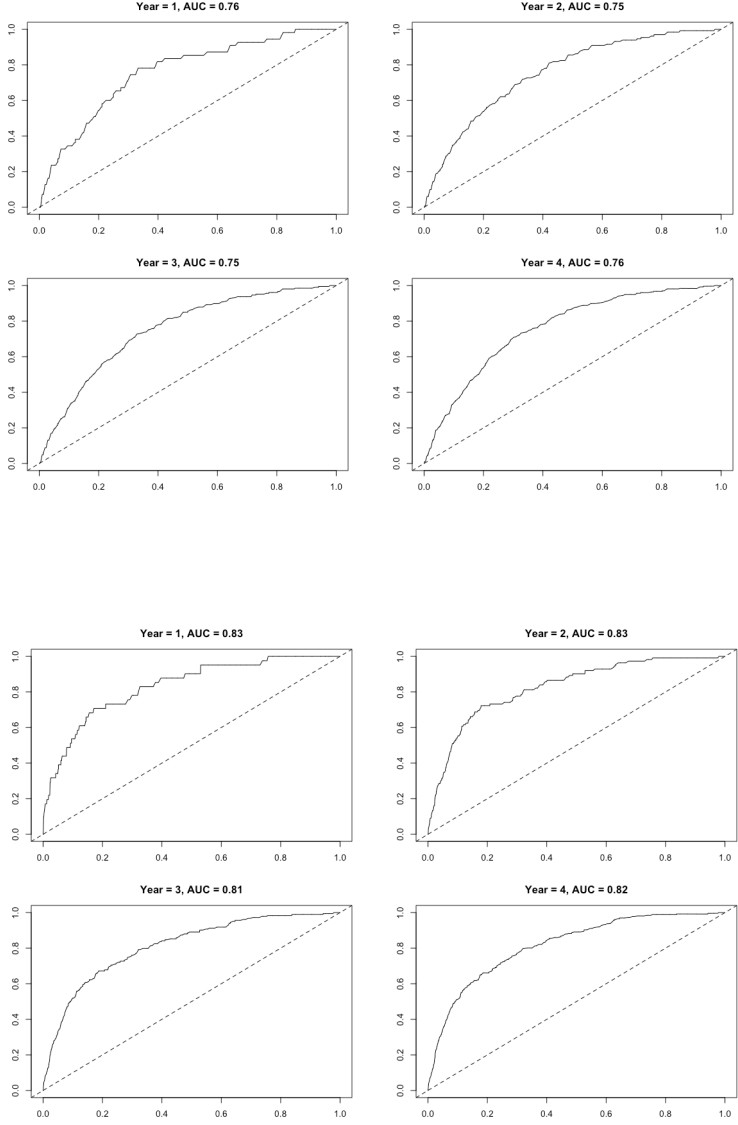

**Figure 1** Receiver operating characteristic (ROC) curves of sensitivity and specificity of the model (work ability predicting mortality risk) within four follow-up years including age and sex (first row columns) and age, sex and work ability (second row columns).

or poor SRH was reported by 35.3% of the participants and 9.8% reported mental health problems (table 1). We identified 275 deaths (1.6% of the participants). The mean follow-up time from the study entry was 3.9 years (SD=0.3 years). Of the participants 2219 (13%) were classified as having limited and 991 (5.8%) poor work ability, and 246 individuals died during the follow-up.

The unadjusted HR for the risk of death among those with limited versus good work ability was 4.58 (95% CI 3.35 to 6.26) and 12.41 (95% CI 9.21 to 16.72) among those with poor versus good work ability. This association

(table 2) was evident across sex and age groups, education levels and participants with and without poor SRH, and with and without mental health problems. However, the association was weaker in men than in women (p value for interaction for poor work ability was 0.04) (table 2). No significant interaction effects between cohorts and work ability were found (p values for interaction with Mini-Finland compared with FinHealth 2017 was 0.21 and with Health 2000 compared with FinHealth 2017 was 0.94). The age and sex adjusted HR for the risk of death among those with poor versus good work ability was ranged from

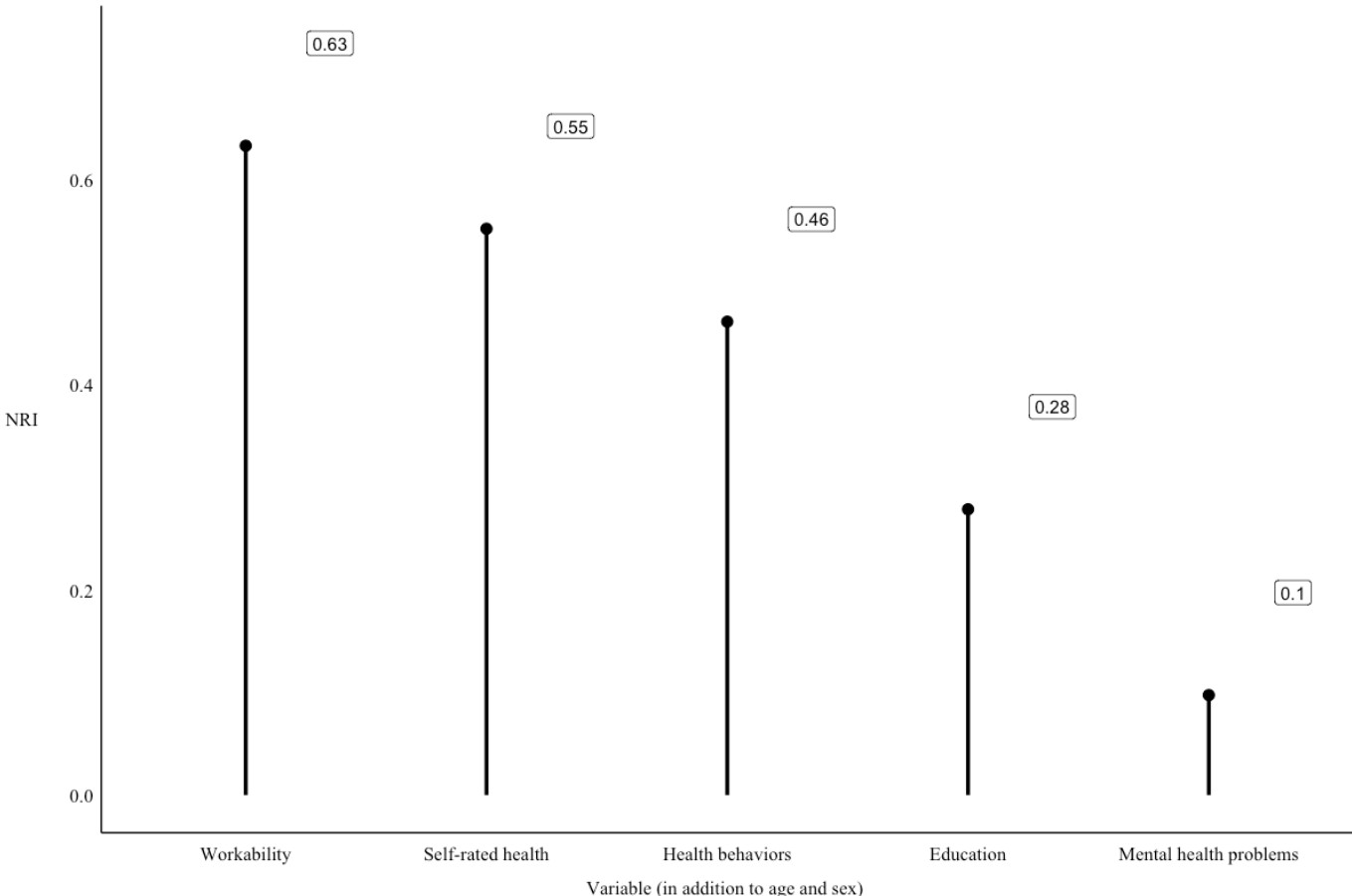

**Figure 2** Net reclassification indexes (NRI) of all included variables predicting mortality risk (the accuracy of the predicted classification of mortality during the 4 year follow-up).

4.12 to 9.8 in individual cohorts (online supplemental figure 1).

Within the combined data, the age and sex adjusted HR for the risk of death among those with limited versus those with good work ability was 3.22 (95% CI 2.30 to 4.53) and 7.20 (95% CI 5.15 to 10.08) among those with poor versus those with good work ability (table 3). The association was robust to further adjustment for education, health behaviours, SRH and mental health problems. In the multivariable analyses (all above-mentioned factors adjusted), the HR for the risk of death among participants with limited versus good work-related functional ability was 3.27 (95% CI 2.12 to 5.05) and 5.75 (95% CI 3.59 to 9.20) among participants with poor versus good work-related functional ability. The latter risk was decreased by 23% compared with the age and sex adjusted association. Health behaviours (PERM 19%) and SRH (PERM 17%) were the strongest contributors to the association.

The ROC curves displaying sensitivity and specificity of our model including (1) age and sex and (1) age, sex and work ability are presented in figure 1. Adding work ability improved the performance of the model and suggested that work ability is a good candidate marker for the mortality risk at all time points. The net reclassification indexes are reported in figure 2. Of all the included variables work ability had the highest NRI. SRH and health

behaviours also increased the accuracy of the predicted classification of mortality during the follow-up.

## DISCUSSION

The present study using three nationally representative samples suggest that individuals with poor work ability are at a higher risk of mortality than are people with good work ability. This association was observed using a simple, one-item measure of work ability with three response categories. The observed fivefold to sixfold excess risk in the pooled data was not explained by socioeconomic and lifestyle risk factors, SRH or mental health problems. The excess risk of mortality in people with poor work ability was also observed in all individual cohorts and thus, our observation may be generalisable over time. Compared with SRH, the association between work ability and mortality risk can be considered as robust and large. Our findings support the notion that SRH and work ability have only partial overlap with each other and represent different dimensions of health and functional ability. Furthermore, work ability measure clearly improved the prediction of our models compared with other predictors including SRH.

It has been shown that poor work ability may predict mortality over a relatively long period of time,[28] but the

majority of the research has been conducted on the associations between self-reported work ability and later disability pension or other forms of exit from work.[29–32] Most of these previous studies used relatively long and multidimensional work ability measures, such as Work Ability Index, although there are studies showing that the predictive validity of one-item measures may be as good as longer and more comprehensive measures.[8]

There are, of course, obvious benefits in measuring work ability with long and comprehensive measures that take into account various medical aspects of the individual's health and functional capacity, the balance between human resources and work demands, the work community, management, the whole psychosocial work environment and environments outside work life.[33] The accuracy and comprehensibility required from the work ability measure depends on the purpose for which it is used, and thus, for example, for granting social benefits the measure of work ability and its influencing factors need to be as valid and reliable as possible. Wide definitions and multidimensional measures are needed specifically when we want to find targets for the interventions aiming to enhance work ability at the individual and organisational level.[7]

However, using comprehensive and extensive measurements in research is not always feasible or even necessary. When studying a wide variety of factors determining population health, healthcare use and healthcare expenditure, it is not possible to measure all potential factors using detailed and long instruments. Thus, short and economic measures with good predictive validity are needed. The one-item work ability measure with three response alternative seems to be a promising option for such a measure, capturing the subjective perception of work ability. This measure is simple, easy to understand and can be used as an indicator of work ability based on the conception people have of their own ability to work. A poor work ability may often be related to multiple factors, including multimorbidity and environmental factors and it is, of course, challenging to reduce such psychosocial and physical risks with multifactorial origin. However, multiple risk-adjusted and stepped-care models that provide access to coordinated care of different levels of intensity have been efficiently applied in occupational medicine to manage work disability.[2 34]

The present study has a number of strengths. To the best of our knowledge, with more than 17 000 participants from three cohorts, the present study is a large and comprehensive examination of the association between work ability and mortality. Information on mortality was obtained from the national health register with a comprehensive recording system for mortality. Thus, the follow-up was virtually complete and independent of active participation in the studies. The Finnish Causes of Death statistics have been reported to be highly reliable.[35] Potential limitations of the study should also be considered. The response rates in individual cohorts ranged from 90% to 68% but the possibility of selection bias in relation to the

investigated exposures and outcomes cannot be totally excluded. Furthermore, the data were limited to participants aged 65 years and younger at baseline. It has been shown that COVID-19 pandemic had a marked impact on all-cause mortality in the European population, after the beginning of 2019 and that excess mortality particularly affected those 65-year-old or older and those with comorbidities.[36] This may have had some small impact on the associations between work ability and mortality in the FinHealth 2017 cohort, but in the pooled data that effect was probably very small. Although all the sample sizes of the individual cohorts included are quite small compared with the whole population it is possible, although highly unlikely that some individuals are included in more than one cohort. However, we do not expect this to be a major source of bias.

Despite these limitations, our findings provide strong population-based evidence to suggest that people with poor self-rated work ability are at increased risk of mortality. Self-rated work ability is a simple and concise measure of health and functional ability with a clear frame of reference and thus it has large potential in screening, epidemiological studies and community-based surveys.

**Contributors** ME, ML, KS and SK contributed to the concept and design of the study. ME performed all data analyses. ME wrote the first draft of the manuscript. ME, ML, KS, A-MA, HR and TJ and SK contributed to the interpretation of the results, manuscript revision and approved the final version of the manuscript. ME is responsible for the overall content as a guarantor.

**Funding** The present study was supported by the Academy of Finland (339390 to ME). The funding bodies had no role in any stage of the research process.

**Competing interests** None declared.

**Patient and public involvement** Patients and/or the public were not involved in the design, or conduct, or reporting, or dissemination plans of this research.

**Patient consent for publication** Consent obtained directly from patient(s).

**Ethics approval** The two latter surveys were approved by the Ethics Committees of the Finnish Institute for Health and Welfare and the Helsinki (NPHI 8/99) and Uusimaa hospital region (HUS 37/13/00/12016), and all participants gave written informed consent. In the 1970's there was no legally prescribed ethical evaluation concerning research work or informed consent procedures, but the general guidelines concerning medicine were also applicable for research. The subjects invited to the Mini-Finland survey were informed on the use of the data for medical research and participation was interpreted as informed consent for research use of the data. Participants gave informed consent to participate in the study before taking part.

**Provenance and peer review** Not commissioned; externally peer reviewed.

**Data availability statement** Data may be obtained from a third party and are not publicly available. Data used in the current study may be obtained from Statistics Finland, and the Finnish Institute for Health and Welfare. Restrictions apply to the availability of these data, which were used under licence for this study. For information on accessing the data see: www.thl.fi.

**ORCID iDs**
Marko Elovainio http://orcid.org/0000-0002-1401-1910
Tuija Jääskeläinen http://orcid.org/0000-0001-6263-2298

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
