## [Reviewer comments · BMJ Open]

ARTICLE DETAILS

TITLE (PROVISIONAL)	Association of short poor work ability -measure with increased mortality risk: a prospective multicohort study
AUTHORS	Elovainio, Marko; Laaksonen, Mikko; Sakari, Kainulainen; Aalto, Anna-Mari; Jääskeläinen, Tuija; Rissanen, Harri; Koskinen, Seppo

VERSION 1 – REVIEW

REVIEWER	M Bethge University of Lübeck, Institute for Social Medicine and Epidemiology
REVIEW RETURNED	07-Sep-2022

GENERAL COMMENTS	The study examined the association between a single-item measure of self-rated work ability and all-cause mortality in three large population-based samples representative of the population of Finland. Work ability was at baseline was assessed by a 3-level measure. The age- and sex-adjusted hazard ratio for mortality risk was 7.20 for participants with poor versus good work ability. The manuscript is easy to read. The methods are appropriate and comprehensible and fully described. I have some minor remarks. My page numbers refer to the page number given in the proof (page x of 30; not original page numbers). Page 6: I wonder if the arguments on self-rated health are also apply to self-rated work ability (e.g. reference group or dependency of other personal factors). Page 17: Please also consider https://pubmed.ncbi.nlm.nih.gov/33219840/ as a reference. Well done!
--

REVIEWER	Antonio Bernabe-Ortiz Universidad Peruana Cayetano Heredia, CRONICAS Centre of Excellence in Chronic Diseases
REVIEW RETURNED	09-Sep-2022

GENERAL COMMENTS	Poor work ability as a risk factor for mortality: evidence from three large population-based cohorts General comment This is a very well-written and interesting paper using pooled data of three population- based cohorts in Finland. Findings can be of interest of BMJ Open readers. Nevertheless, there are some concerns that need to be clarified. Of note, given the results, work ability can be a good predictor of mortality, but seems to be merely anecdotic. A poor work ability may be related to multiple factors, including multimorbidity; so, an issue to discuss could be related to how to reduce such risk. Some lines should be added in the
--

	discussion section. Specific comments The abstract is not clear in many aspects. Add between brackets “good work ability” as the completely fit category of work ability. Limited work ability should be included as an important result. There is no information about considering work ability as predictor as in the manuscript. Thus, a reader can think that only association has been evaluated, and not prediction as established in the conclusions. Please add a subheading for the Introduction. After reading this section, it seems the paper will focus on a comparison between the prediction of SRH and work ability for all- cause mortality, but the aim/hypothesis is not in that sense. Maybe to add a secondary aim may help here, especially because some results are focus on such comparison. How the three cohorts’ data was combined needs to be explained, especially because they have different framework approach. What is the possibility that one participant was in two of the cohorts? Was that considered? Why the follow-up time was restricted to four years? The first paragraph of the work ability subheading seems to be incomplete. Please verify. Regarding mortality: How time until death was collected? This need to be clear for Cox analysis. Please add a brief description of the quality of the registries used. How about the impact of the COVID pandemic in all-cause mortality rates? A brief explanation is needed as this may impact on results from the FinHealth 2017 cohort. In the discussion section, please add some lines regarding how to reduce the high risk of mortality in participants with poor work ability. Please verify the title of Figure 1. Something is not clear there (lower part).
--	--

VERSION 1 – AUTHOR RESPONSE

Reviewer: 1

The study examined the association between a single-item measure of self-rated work ability and all-cause mortality in three large population-based samples representative of the population of Finland. Work ability was at baseline was assessed by a 3-level measure. The age- and sex-adjusted hazard ratio for mortality risk was 7.20 for participants with poor versus good work ability. The manuscript is easy to read. The methods are appropriate and comprehensible and fully described.

Our response: We appreciate the positive feedback.

I have some minor remarks. My page numbers refer to the page number given in the proof (page x of 30; not original page numbers). Page 6: I wonder if the arguments on self-rated health are also apply to self-rated work ability (e.g. reference group or dependency of other personal factors).

Our response: We understand the point, removed the justification related to self-rated health and have now rewritten the introduction as follows: “In the rapidly ageing working population work ability is an important dimension of overall health and maintaining work ability is important for countries seeking to keep their citizens in work much longer than previous generations (1). The rapid changes

in working life over the past few decades, with a shift from physical demands towards cognitive and psychosocial demands, together with changes in the functional capacity, expertise, and attitudes of workers, is reflected in the content of work ability. Poor work ability represents a considerable challenge to the sufficiency of the workforce and to the sustainability of social protection schemes. Promoting the work ability of the working-age population helps people to maintain their health and functional capacity enabling longer working careers and may also improve the quality of life of the growing population of retirees.

There are efficient risk-adjusted, stepped care models and interventions for work disability management (2), but these models and interventions need to be focused on those that really need and benefit from them. The primary precondition for correctly focused interventions is easily administered and reliable measure of work disability. Currently the identification of workers in increased risk of work disability is based on the duration of sickness absence (3) Sickness absence is a good predictor of ill health and permanent work disability (4), but only a small proportion of those with even long -term sickness absences really need sustained support to maintain work ability (5). Thus, there is a clear need of a simple, easy to administer and unambiguous measures of work-related health and functional ability that can predict severe health outcomes and health service use. Such measures are also essential for health surveys and population health monitoring.

One of the most widely used one is the Work Ability Index, a self-report survey screening instrument (6, 7). The questionnaire assesses with 7 dimensions and 16 questions the degree to which workers consider their state of health adequate to cope with their job demands. The continuous scores can be categorized into groups that reflect different levels of need of support. Although widely used and reliable (1), the work ability index is still rather complicated and inconvenient for use in large-scale surveys (8). A more simple, single-item question on self-rated health (SRH) has been commonly used as a measure of health status in psychological research, clinical settings, and general population surveys. It has also repeatedly been shown that SRH associates with physicians' assessments of health (9, 10), chronic disease incidence, physical and cognitive functional limitations, health services use, clinical biomarkers (11-14) and even mortality (15-17).

In this study, we tested a self-administered one-item measure of work ability as a predictor of overall mortality. The subjective experience of work ability has the advantage over general health evaluation that it has a clear point of reference: subjective work ability is a measure of perceived balance of the demands of the work and the resources of the individual (18). Although the conceptual framework is, of course, much more complicated, the evaluation of ones' capacity to cope with the demands of one's work is probably much less abstract and cognitively challenging than evaluating the overall health status. We tested whether subjective evaluation of work ability predicts the risk of overall mortality in three large and nationally representative cohorts collected in the 1970s, 2000s and 2010s. We also tested whether the one-item work ability measure predict mortality risk as well as and independently of the one-item self-rated health measure."

Page 17: Please also consider <https://pubmed.ncbi.nlm.nih.gov/33219840/> as a reference.

Our response: That was an excellent suggestion and have now added the reference (Bethge M, Spanier K, Kohn S, et al. Self-reported work ability predicts health-related exit and absence from work, work participation, and death: longitudinal findings from a sample of German employees. *Int Arch Occup Environ Health* 2021;94(4):591-9.). In the text: "In the rapidly ageing working population work ability is an important dimension of overall health and maintaining work ability is important for countries seeking to keep their citizens in work much longer than previous generations (1)."

And

"Although widely used and reliable (1), the work ability index is still rather complicated and inconvenient for use in large-scale surveys (8)."

Reviewer: 2

Comment 1:

This is a very well-written and interesting paper using pooled data of three population-based cohorts in Finland. Findings can be of interest of BMJ Open readers. Nevertheless, there are some concerns that need to be clarified. Of note, given the results, work ability can be a good predictor of mortality, but seems to be merely anecdotic. A poor work ability may be related to multiple factors, including multimorbidity; so, an issue to discuss could be related to how to reduce such risk. Some lines should be added in the discussion section.

Our response: We have now added some lines about the multifactorial origin of work disability and the problems related to reducing such risks. In the text: "A poor work ability may often be related to multiple factors, including multimorbidity and environmental factors and it is, of course, challenging to reduce such psychosocial and physical risks with multifactorial origin. However, multiple risk-adjusted and stepped-care models that provide access to coordinated care of different levels of intensity have been efficiently applied in occupational medicine to manage work disability (2, 34)."

Comment 2:

The abstract is not clear in many aspects. Add between brackets "good work ability" as the completely fit category of work ability. Limited work ability should be included as an important result. There is no information about considering work ability as predictor as in the manuscript. Thus, a reader can think that only association has been evaluated, and not prediction as established in the conclusions.

Our response: We apologise for being unclear and we have now added "good work ability" between brackets as the completely fit category of work ability. We have included the results of limited work ability and clarified the predictive role of work ability.

In the abstract: "Objectives: To examine whether a single-item measure of self-rated work ability predict all-cause mortality in three large population-based samples collected in 1978-1980, 2000 and 2017." And ... and 3.22 (95% CI 2.30–4.43) for participants with limited versus good work ability."

And in the text: "We tested whether subjective evaluation of work ability predicts the risk of overall mortality in three large and nationally representative cohorts collected in the 1970s, 2000s and 2010s. We also tested whether the one-item work ability measure predict mortality risk as well as and independently of the one-item self-rated health measure."

Comment 3: Please add a subheading for the Introduction.

Our response: Subheading is now added as suggested.

Comment 4: After reading this section, it seems the paper will focus on a comparison between the prediction of SRH and work ability for all-cause mortality, but the aim/hypothesis is not in that sense. Maybe to add a secondary aim may help here, especially because some results are focus on such comparison.

Our response: We have now added a secondary aim as suggested. In the text: "We also tested whether the one-item work ability measure predict mortality risk as well as and independently of the one-item self-rated health measure."

How the three cohorts' data was combined needs to be explained, especially because they have different framework approach. What is the possibility that one participant was in two of the cohorts? Was that considered? Why the follow-up time was restricted to four years?

Our response: All data sets were sampled to represent the Finnish population aged over 30 ("Mini-Finland health survey", "Health2000) or over 18 years ("FinHealt 2017"). As the population of Finland is approximately 5 m. and final sample 17 000, it is possible, although not very probable that some individuals are included in more than one data set. The follow-up time was restricted to four years because that was the maximum follow-up time in the latest cohort. This is now also reported in the methods section as follows: "For the main analyses we combined these three data sets. We restricted the follow-up time to maximum 4 years, because that was the maximum follow up time in the latest (FinHealth2017) cohort to make data sets comparable."

We also discussed the possibility of overlap as a potential limitation of the study. In the text: "Although all the sample sizes of the individual cohorts included are quite small compared the whole population it is possible, although highly unlikely that some individuals are included in more than one cohort. However, we do not expect this to be a major source of bias."

The first paragraph of the work ability subheading seems to be incomplete. Please verify.

Our response: We apologise, there was still text from a previous version of the manuscript. That is now corrected and reads: "In all cohorts, self-rated work ability was assessed at baseline (1978-80, 2000-2001 and 2017) using a single question: "Regardless of whether you are employed or not, please estimate your current work capacity. Are you?" The response alternatives were "completely fit for work", "partially unable to work" and "completely unable to work". These categories will be referred as good, limited and poor, respectively."

Regarding mortality: How time until death was collected? This need to be clear for Cox analysis.

Our response: The deaths and the date of deaths were from the Causes of Death registry including information of all deaths in Finland. This is now clarified in the text as follows: "Information on deaths and the dates of death were obtained from the Causes of Death registry maintained by Statistics Finland and the Death registry from the Digital and population data services agency until the end of 2020. The Causes of Death registry includes information of all deaths in Finland."

Please add a brief description of the quality of the registries used.

Our response: We have now added this information and the reference. In the text: "Information on mortality was obtained from the national health register with comprehensive recording system for mortality. Thus, the follow-up was virtually complete and independent of active participation in the studies. The Finnish Causes of Death statistics have been reported to be highly reliable (35)."

How about the impact of the COVID pandemic in all-cause mortality rates? A brief explanation is needed as this may impact on results from the FinHealth 2017 cohort.

Our response: We added a brief explanation in the discussion as follows: "It has been shown that COVID-19 pandemic had a marked impact on all-cause mortality in the European population, after the beginning of 2019 and that excess mortality particularly affected those 65-year-old or older and those with co-morbidities (36). This may have had some small impact on the associations between work

ability and mortality in the FinHealth 2017 cohort, but the in the pooled data that effect was probably very small.”

In the discussion section, please add some lines regarding how to reduce the high risk of mortality in participants with poor work ability.

Our response: We have added discussion on the issue. In the text: “A poor work ability may often be related to multiple factors, including multimorbidity and environmental factors and it is, of course, challenging to reduce such psychosocial and physical risks with multifactorial origin. However, multiple risk-adjusted and stepped-care models that provide access to coordinated care of different levels of intensity have been efficiently applied in occupational medicine to manage work disability (2, 34).”

Please verify the title of Figure 1. Something is not clear there (lower part).

Our response: The “lower part” is now changed into “second row columns”.

VERSION 2 – REVIEW

REVIEWER	Antonio Bernabe-Ortiz Universidad Peruana Cayetano Heredia, CRONICAS Centre of Excellence in Chronic Diseases
REVIEW RETURNED	07-Dec-2022

GENERAL COMMENTS	Page 9, line 44: Correct the word "lates" for "latest".
---